# Towards Scientific Data Synthesis Using Deep Learning and Semantic Web

Alsayed Algergawy [iD] , Hamdi Hamed[iD] , and Birgitta König-Ries[iD]

Heinz-Nixdorf Chair for Distributed Information Systems,
Institute for Computer Science, University of Jena, Germany
{alsayed.algergawy|hamdi.hamed|birgitta.koenig-ries@uni-jena.de}

**Abstract** One of the added values of long running and large scale collaborative projects is the ability to answer complex research questions based on the comprehensive set of data provided by their central repositories. In practice, however, finding data in such a repository to answer a specific question often proves to be a demanding task even for project scientists. In this paper, we aim to ease this task, thereby enabling cross-cutting analyses. To achieve that we introduce a new data analysis and summarization approach combining semantic web and machine learning approaches. In particular, the proposed approach makes use of the capability of machine learning to categorize a given dataset into a domain topic and to extract hidden links between its data attributes and data attributes from other datasets. The proposed approach has been developed in the frame of CRC AquaDiva[1]and has been applied to its datasets.

## 1 Introduction

The Collaborative Research Center (CRC) AquaDiva is a large collaborative project spanning a variety of domains including biology, geology, chemistry, and computer science with the common goal to better understand the Earth's critical zone. Datasets collected within AquaDiva, like those of many other cross-institutional, cross-domain research projects, are complex and difficult to reuse since they are highly diverse and heterogeneous [2,10]. This limits the dataset accessibility to the few people who were either involved in creating the datasets or have spent a significant amount of time aiming to understand them. This is even true for scientists working in other parts of the same project. They, too, will need time to figure out the major theme of unfamiliar datasets. We believe that dataset analysis and summarization can be used as an elegant way to provide a concise overview of an entire dataset. This makes it possible to explore in depth datasets of potential interest, only [7,8,9].

Most previous work on data summarization focuses on text summarization [4], while few works only are pivoted to summarizing tabular data [7,9]. Most of them focus on schema summarization, which provides an

---

[1] http://www.aquadiva.uni-jena.de/

overview of a schema with a set of representative attributes [8]. Even though there are a number of approaches creating a summarization of instances of a table instead of schema elements [6,7], they ignore the context of the dataset, i.e., they deal with the dataset independent from other datasets in the same data repository. This makes it difficult to find related datasets. In contrast, we aim to summarize and link datasets thereby enabling cross-cutting analyses. To this end, we develop an approach that semantically classifies data attributes of scientific datasets (tabular data) based on a combination of semantic web and deep learning. This classification contributes to summarizing individual datasets, but also to linking them to others. We view this as an important building block for a larger data summarization system. We believe that figuring out the subject of a dataset is an important and basic step in data summarization. To this end, the proposed approach categorizes a given dataset into a domain topic. With this topic, we then extract hidden links between different datasets in the repository. The proposed approach has two main phases: 1) *off-line* to train and build a classification model using a supervised deep learning using convolution layers and 2) *on-line* making use of the pre-trained model to classify datasets into the learned categories. With this, we (i) allocate the dataset into a set of dataset topics called *domain signatures*, and (ii) locate hidden links between the dataset and other datasets in the same data repository. To demonstrate the applicability of the approach, we analyzed and synthesized datasets from the AquaDiva Data Portal [2].

## 2 Proposed Approach

Over the last few years, convolutional networks have been successfully used for numerous classification tasks. It thus seemed natural to explore whether they could be useful in our dataset classification task, too. We faced two main challenges: First, we needed to decide which information contained in a dataset was most likely to contribute to the classification. The second was how to encode this information in such a way that on the one hand it would be suitable input to a CNN and on the other hand could be done with reasonable effort.

To address the first challenge, let us take a closer look at datasets: A dataset is defined as a tuple of primary data and metadata organized for a specific purpose. Each tuple in the primary data is a collection of data cells containing data values (called *data points*) [1]. The metadata contains information about, e.g., the data owner, data curators, the methodology used to produce primary data, etc. The primary data represents the actual data organized according to a specific structure, called *data structure*. Each data structure consists of a set of data attributes, each data attribute has a name, datatype, (optional) unit, description, as well as a semantic annotation based on a domain ontology. In the context of dataset analysis and summarization, this annotation is an important basis. Consider as an example, two datasets $\mathcal{DS}_1$ and $\mathcal{DS}_2$ stored in the AquaDiva data repository. Almost all data attributes

---
[2] https://addp.uni-jena.de/

of available datasets in that repository are annotated using the AquaDiva ontology ($ADOn$) as the domain specific ontology. The first dataset $\mathcal{DS}_1$ contains `weather and soil monitoring` data. It has a data structure with 50 data attributes including *"soil temperature"* annotated with the concepts soil (`http://purl.obolibrary.org/obo/ENVO_00001998`) and characteristic temperature (`http://purl.obolibrary.org/obo/PATO_0000146`). The second dataset $\mathcal{DS}_2$ provides information about `soil moisture` in the Hainich forest. $\mathcal{DS}_2$ has a data structure with 13 data attributes. The *"mean_theta_forestbottom"* data attribute is also annotated with the concept soil and the characteristic soil moisture (`http://www.aquadiva.uni-jena.de/ad-ontology/ad-ontology.0.0/ad-ontology-characteristics.owl#SoilMoisture`). Analysing the dataset annotation, a possible relationship of the two datasets $\mathcal{DS}_1$ and $\mathcal{DS}_2$ can be determined. Based on these considerations, we argue that data attributes are the most important parts of the dataset and the data preparation process will take place around them. Furthermore, metadata provides an important source for understanding and interpreting datasets. Therefore, constructing a new structure via data preparation is mainly based on data attributes and metadata. We gathered all information related to data attributes, where for each data attribute we consider *name, datatype, unit, data points* attached to the data attribute as well as its *semantic annotation*. Furthermore, we use the dataset title contained in the metadata as textual representation of the dataset.

With respect to the second challenge, we took the following approach: Convolutional networks are fundamentally geared towards image data, (i.e., matrices with size $n \times m$) and perform feature extraction and classification via hidden layers [5]. We therefore decided to experiment with a very straightforward encoding. We basically take "photos" of relevant parts of the dataset containing the information described above. More precisely, an image is generated for each data attribute. Figure 1 illustrates the "*Air temperature_mean*" data attributes from the "*Weather and soil data monitoring*" along with its annotation, data type (decimal), unit (Celsius), and 30 data points.

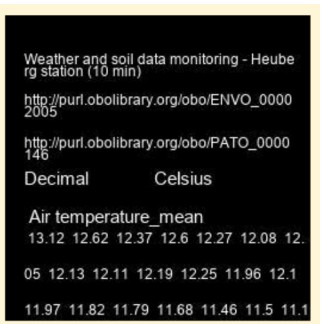

**Figure 1:** Image content

Together, this results in the approach shown in Fig. 2, with the *model building (training)* and *model deployment (operating)* phases. In the following, we describe the main steps of each phase.

**Data preparation.** The main objective of this component is to prepare the large number of heterogeneous datasets for analysis. It is needed both during the model building (*training phase*) and the model deployment (*operating phase*) to convert each dataset into a structure suitable for the next component, *image generation*. To this end, we propose

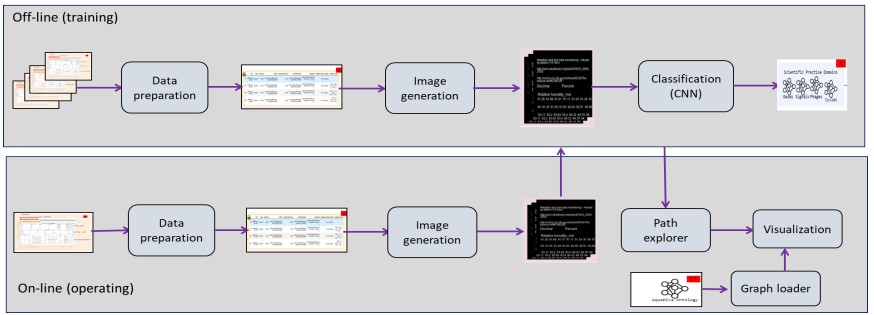

**Figure 2:** Framework architecture

a new structure that combines the selected features described above from the dataset into a single container.

**Image generation.** This components transforms the created structure in a series of images as described above.

**Classification.** In this step, we aim to cluster dataset data attributes according to the domain signatures (collected and defined by domain experts) that the dataset relates to. It applies a set of convolutional layers over the input images representing dataset characteristics over the mentioned dimensions, to extract the hidden features from the textual representations of the 5 dimensions. The classification outputs the top 5 classifications results, corresponding to the top 5 domain signatures the dataset relates to in the data repository. In the current implementation, we use the ResNet18 convolutional neural network[3] to build the classification model, as it achieved the best results in our trials.

**Model-Deployment (Operating) phase**. Once we have build the classification model, the system is ready to analyze new datasets, as shown in Fig. 2. A first step is to prepare the input dataset through the *data preparation* component and to generate the corresponding set of images via the *image generation* component. These images are used as input to the pre-trained model to get the correct scientific domain signature. The outcome of the classification model along with the domain specific ontology are used to generate the domain signature of the given dataset and also to link its data attributes to corresponding data attributes from other datasets in the same data repository. The results are then visualized using the *visualization* component, as shown in Fig. 2. A screen shot of the result is illustrated in Fig. 3. The result of this analysis illustrates that the main topic of this dataset is "site and water quality". The figure also shows that the dataset is linked to a number of other datasets, e.g. datasets 92, 128 and 214 via the concept "groundwater" as shown in Fig. 3.

We build and test the proposed approach using datasets of the AquaDiva data portal[3]. We used 114 datasets representing different domain topics, such as weather monitoring, groundwater hydrochemistry, gene abundance, or soil physical parameters. 70% of these were used for training and 30% for evaluation.

---

[3] due to the data policy of the CRC, we can not publish these datasets.

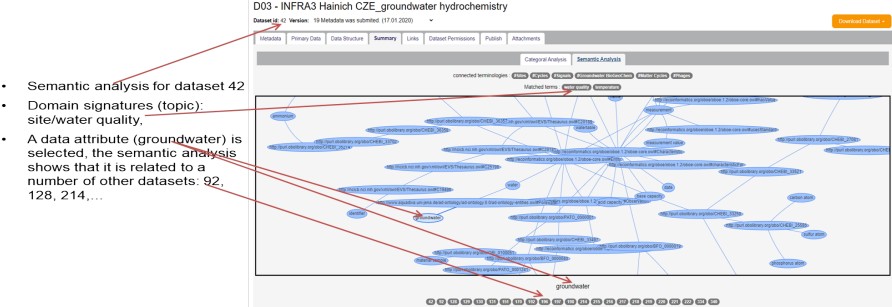

**Figure 3:** Semantic linking visualization

The total number of data attributes is 1300; the number of data points within a dataset ranges from 300 (5 data attributes × 60 tuples) to 12,000,000. Example results were presented to domain experts from CRC AquaDiva. They confirmed the correctness and usefulness of this classification. Based on these promising first results, we aim to extend and further evaluate the approach in future work.

All resources related to the proposed approach as well as the preliminary results are accessible at `https://github.com/fusion-jena/JeDaSS`.

**Acknowledgments:** This work has been funded by the *Deutsche Forschungsgemeinschaft* (CRC AquaDiva, Project 218627073).

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
