# OpenReview forum: "Towards Scientific Data Synthesis Using Deep Learning and Semantic Web"
_eswc-conferences.org/ESWC/2021/Conference/Poster_and_Demo_Track — ESWC2021 P&D_

### Official Review · AnonReviewer3 · 2021-04-07
**Not very relevant to ESWC**

**Rating:** 5
**Confidence:** 3

**Review:**

This paper presents a table classification system. I am not sure whether the paper is relevant to ESWC. An ontology is used as a topic hierarchy for classification. All other techniques and applications are not very relevant to the Semantic Web.

Regarding the techniques, some important details are missing. The key step in the proposed approach seems the transformation of a structure into a set of images, but it is not clear how the transformation is implemented.

It is also not clear how to link attributes to other datasets. Is it also formulated as a classification problem?

**Anonymity:**

Yes, I would like my review to remain anonymous.

---

### Official Review · AnonReviewer1 · 2021-04-14
**Dataset classification with major flaws**

**Rating:** 3
**Confidence:** 4

**Review:**

Review
The authors propose an approach to classify tabular datasets based on the given dataset relations (consisting of a name, type, unit, data points and connections to an ontology) and the title of the datasets. The classification attributes are combined into multiple matrices, which are classified by a ResNet18 CNN architecture. The classification results are used to connect datasets and visualize these connections. The approach is evaluated by domain experts based on one example dataset.

# Strengths
* Classifying scientific datasets form a larger collection is a promising direction to improve the findability and reusability, especially in interdisciplinary research.
*The AquaDiva project seems to be a good resource for this use case, because understanding the earth's Critical Zone appears to be an interdisciplinary research field with a large potentially of reusing the generated datasets in other fields.

# Weaknesses
* The paper provides only a small contribution. The used classifier is not particularly recent and no justification is given why this classifier should work better than any other approach on the given data.
* Many important details are omitted from this paper. The normalization of the classification attributes, which would be required to use a CNN is not explained at all nor are relevant details about the classifier given (e.g. number of classes, used training data).
* The paper starts with discussing dataset summarization but works on dataset classification without sufficiently showing the connection between these topics.

# Detailed Comments
* There is a discrepancy between the abstract, introduction and the proposed classification. Initially, dataset summarization is motivated and related work about this field is discussed, but the actual contribution of this paper is in classification. If classification is seen as an intermediate step towards summarization that should be explained in detail.
* The explanation of the data structure is not well organized. Beside introducing the names of the classification attributes only a limited amount of information is given. What are data points? What are the annotations based on the AquaDiva ontology? Here it would be useful to introduce an example.
* No statistics about the used collection are presented. How many datasets are in this collection? How many dataset attributes and data points are in one dataset? Is the metadata available for all datasets? How much training data was used to train the classifier?
* The preprocessing step could limit the attribute selection to a smaller description and add a detailed description on how these heterogeneous attributes are transformed into matrices/images. It is unclear how the textual data is preprocessed. What features are used for? TF-IDF? Word embeddings? Based on this missing information it is also not clear why a CNN should be able to extract / recognize patterns in the given matrices.
* No information about the classifier is given and the two details given contradict each other. If the final layer of this classifier has 5 dimensions this classifier would be able to classify 5 classes, but the paper states that the top 5 results are given for each dataset, which indicates that there are more than 5 classes. Additionally, the provided example shows only two topics/ classes.

## Minor Comments
* The abstract mentions CRC before this acronym is defined in  the Introduction.
* “[...] and difficult to reuse since they they are highly diverse and heterogeneous” word repetition of they in the introduction.
* “It aims to cluster datasets” in the classification paragraph might lead to misunderstandings about the used method (clustering or classification). It would be better to use “to categorize” or “to classify” to avoid this.
* Pre-training most commonly refers to the training of the model on a different task and/or dataset, like word embeddings, which are pre-trained on the cloze task, before using them in all kinds of architectures. The proposed “pre-training” seems to be the simple training of the CNN on the exact task.
* For this paper it would be better to talk about matrices instead of images. Images seems to be not the right term here.
* The basic workflow of a classifier is described two times. It can be assumed that the reader knows that a classifier is trained on specific data and later on used to predict the classes. Removing this explanation would save space without hurting the general understandability of this paper.

Overall, the paper presents an interesting idea to improve the findability and sequentially enables better (re-)use of the existing dataset resources by using classification. However, the classifier is not described in detail nor is a ResNet18 CNN a particular recent classifier, which would promise better results than other classifiers. The paper does not describe how the heterogeneous data (textual data, data points, type information, etc.) is transformed into matrices. Additionally, the idea of using a CNN to extract local patterns out of the created matrices is not justified. Combined with the discrepancy between the abstract and introduction focusing on dataset summarization and the actual contribution in the field of dataset classification these points lead to the recommendation to not publish this paper in its current state.


**Anonymity:**

Yes, I would like my review to remain anonymous.

---

### Official Review · AnonReviewer2 · 2021-04-15
**Interesting application combining DL and SW with real datasets and a tool in use by domain experts**

**Rating:** 9
**Confidence:** 4

**Review:**

This paper proposes an approach to help summarize and analyze a large number of heterogeneous datasets from the AquDiva. The authors (1) train a CNN to learn features from datasets to classify them using domain expert-assigned topics, and (2) deploy the model to provide prediction outputs to a visualization tool for analyzing datasets with positive feedback from domain experts.

Reasons to accept:
- An interesting application that
    + analyzes real datasets from a collaborative project
    + uses both machine learning and semantic web for training the model
    + has a visualization tool leveraging predictions from the trained model to be used by domain experts
    + has positive feedback from domain expert users


**Anonymity:**

Yes, I would like my review to remain anonymous.

---

### Official Review · ~Aidan_Hogan1 · 2021-04-18
**Metareview: Accept (But requires further details and clarifications)**

**Rating:** 6
**Confidence:** 5

**Review:**

This was a controversial paper on which there were strong opinions on either side. Overall the reviews tend to agree that the application is of interest, and it is particularly commendable that the system is in-use with domain experts. In terms of reasons to reject, certain important details are omitted, and one reviewer found the connection to the Semantic Web unclear. In the end, we have decided to accept the paper per the fact that it is an interesting application, but with the condition that the authors revise the paper to clarify the relation to the Semantic Web, and add the additional details requested by the reviewers, potentially in an accompanying webpage or extended version linked from the paper if there is not enough space in the paper.

**Anonymity:**

No, I would like my review to be deanonymized.

---

### Decision · Program_Chairs · 2021-04-19

Accept